# Casein Kinase I Protein Hrr25 Is Required for Pin4 Phosphorylation and Mediates Cell Wall Integrity Signaling in *Saccharomyces cerevisiae*

**DOI:** 10.3390/genes16010094

**Published:** 2025-01-17

**Authors:** Amita Bhattarai, Manika Bhondeley, Zhengchang Liu

**Affiliations:** 1Department of Biological Sciences, University of New Orleans, New Orleans, LA 70148, USA; 2CTC Core Laboratory, Tulane University School of Medicine, New Orleans, LA 70112, USA; 3Kudo Biotechnology, 117 Kendrick Street, Needham, MA 02494, USA

**Keywords:** casein kinase I, Hrr25, Pin4, Slt2, Bck1, Rlm1, cell wall integrity, protein phosphorylation

## Abstract

Background: Casein kinase I protein Hrr25 plays important roles in many cellular processes, including autophagy, vesicular trafficking, ribosome biogenesis, mitochondrial biogenesis, and the DNA damage response in *Saccharomyces cerevisiae*. Pin4 is a multi-phosphorylated protein that has been reported to be involved in the cell wall integrity (CWI) pathway and DNA damage response. Pin4 was reported to interact with Hrr25 in yeast two-hybrid and large-scale pulldown assays. Methods/Objectives: Co-immunoprecipitation and yeast two-hybrid assays were utilized to confirm whether Pin4 and Hrr25 interact and to determine how they interact. Genetic interaction analysis was conducted to examine whether *hrr25* mutations form synthetic growth defects with mutations in genes involved in CWI signaling. Immunoblotting was used to determine whether Hrr25 phosphorylates Pin4. Results: We show that Hrr25 interacts with Pin4 and is required for Pin4 phosphorylation. *pin4* mutations result in synthetic slow-growth phenotypes with mutations in genes encoding Bck1 and Slt2, two of the protein kinases in the MAP kinase cascade that regulates CWI in the budding yeast. We show that *hrr25* mutations result in similar phenotypes to *pin4* mutations. Hrr25 consists of an N-terminal kinase domain, a middle region, and a C-terminal proline/glutamine-rich domain. The function of the C-terminal P/Q-rich domain of Hrr25 has been elusive. We found that the C-terminal region of Hrr25 is required both for Pin4 interaction and CWI. Conclusions: Our data suggest that Hrr25 is implicated in cell wall integrity signaling via its association with Pin4.

## 1. Introduction

In *Saccharomyces cerevisiae*, the cell wall provides resistance to osmotic shock and mechanical stress, determines the cell shape, and serves as a surface for peripheral glycoproteins [1,2,3,4]. Thus, cell wall integrity maintenance is crucial during normal cell growth and environmental stress conditions. Cell Wall Integrity (CWI) signaling is the central pathway that is employed by yeast cells to respond to cell wall perturbations [5,6]. CWI signaling is conserved in the fungal kingdom, including several clinically important human fungal pathogens [7]. Multiple antifungals target the synthesis of the cell wall components, for example, β-glucan, chitin, and mannoproteins [8]. Mechanistic insights into the cell wall synthesis and regulation of CWI signaling could be key to improving existent or developing new antifungals.

Cell wall stress conditions are first recognized by cell-surface mechanosensory receptors Wsc1/2/3, Mid2, and Mtl1. The signal then travels through the pathway to result in a transcriptional response. A physical interaction of a Ras-like, GTP-bound protein Rho1 with a serine/threonine protein kinase C (Pkc1) initiates the hierarchical CWI mitogen-activated protein (MAP) kinase cascade. The CWI MAP kinase cascade consists of a MEK kinase Bck1, two homologous MEKs, Mkk1 and Mkk2, and a MAP kinase Slt2 (Mpk1). The kinases are sequentially activated one after another. Once phosphorylated, Slt2 initiates the transcription of genes involved in cell wall homeostasis through the serum response factor–like transcription factor Rlm1 and the Swi4/6 SBF transcription complex [9,10,11,12]. Mlp1, a pseudokinase paralog of Mpk1, also contributes to the regulation of Rlm1 and Swi4/6 through a noncatalytic mechanism [9,11].

Apart from well-described cell wall stressors, such as Calcofluor white, Congo red, zymolyase, and caffeine, the CWI pathway is activated in response to many other factors: heat and osmotic stress, exposure to mating pheromones, actin cytoskeleton depolarization, and ER stress [6]. These findings have led to the proposal that stressors provide “lateral” inputs and converge onto different components of the central CWI MAP kinase cascade, a concept that helps to explain the sophisticated cellular responses to a diverse range of CWI pathway activators.

The poorly studied phosphoprotein Pin4 (Mdt1) has been shown to be involved in DNA damage response and CWI maintenance; however, the upstream kinase regulating Pin4 in the latter process has not been reported [13]. Pin4 has been found to interact with Hrr25 in several large-scale protein complex pull-down assays [14,15,16]. Recently, Murakami-Sekimata et al. reported that Hrr25 and Pin4 interacted in a yeast two-hybrid assay, but the authors noted that they were unable to show the physical binding of Pin4 to Hrr25 [17]. Nevertheless, their genetic analysis revealed that *pin4*Δ and an *hrr25* missense mutation, *hrr25(T176I)*, suppressed the temperature-sensitive growth phenotype of a *sec12-4* mutant, which is defective in vesicle budding from the endoplasmic membrane in the protein section pathway [17,18]. In this report, we extend the analysis of the interaction between Hrr25 and Pin4, having found that Hrr25 is required for Pin4 phosphorylation. We further show that Hrr25 interacts with Pin4 through its C-terminal proline/glutamine-rich region, attributing for the first time, to our knowledge, a function to that domain. By analyzing the phenotypes of *hrr25* mutations in combination with *bck1* and *slt2* mutations, we show that Hrr25 is also involved in cell wall integrity signaling, consistent with the reported role of Pin4 in this pathway. We propose that Pin4 and Hrr25 form a new branch in cell wall integrity sensing and maintenance.

## 2. Materials and Methods

### 2.1. Strains, Plasmids, Growth Media, Growth Conditions, and Yeast Transformation

The yeast strains and plasmids used in this study are listed in Table 1 and Table 2, respectively. Yeast strains were grown in YPD (1% Bacto Yeast Extract, 2% Bacto Peptone, 2% glucose) (Fisher Scientific, Waltham, MA, USA), YNBcasD (0.67% yeast nitrogen base, 0.25% casamino acids, 2% dextrose) (Fisher Scientific), YNBcas5D (similar to YNBcasD, with 5% dextrose), and complete supplement mixture medium (CSM) (0.67% yeast nitrogen base, 2% glucose, 0.6 g/L CSM minus histidine, leucine and tryptophan (Sunrise Science Products, Inc., Knoxville, TN, USA)), as indicated in the text or in the figure legends. Caffeine (Fisher Scientific) was filter sterilized and added to YPD medium after it was autoclaved and cooled down to about 60 °C. Sorbitol (United States Biological, Salem, MA, USA) was added at 1 M concentration to YPD medium before autoclaving. Agar at a concentration of 2% (United States Biological) was added to make solid growth medium. Cells cultured in liquid medium were grown in a shaking incubator at 220 rpm at 30 °C. Plate-grown cells were incubated at 30 °C. Yeast strains were transformed using the high-efficiency method [19]. Gene deletion mutants carrying the *kanMX4* and *HIS3MX6* disruption cassettes were selected on YPD medium supplemented with 300 mg/L geneticin (United States Biological) and CSM medium without histidine, respectively. Yeast strains transformed with pRS416-derived vectors were selected on YNBcasD medium. Deletion mutant strains were generated and confirmed as described [20].

### 2.2. Yeast Two-Hybrid Analysis

For yeast two-hybrid analysis, the *PIN4* coding sequence was amplified by PCR and cloned into pRS415-TEF2-GAD, a Gal4 transcriptional activation domain vector (GAD), to generate the pRS415-TEF2-GAD-PIN4 plasmid. Plasmids encoding Gal4-DNA binding domain with wild-type Hrr25, kinase-dead mutant Hrr25(K38A), and various Hrr25 fragments were generated, as described previously [23]. Yeast two-hybrid strains AH109 and Y187 (Takara Bio USA, Inc., San Jose, CA, USA) were transformed with plasmids expressing GBD and GAD constructs, respectively. AH109 and Y187 transformants were mated to generate diploid strains coexpressing GBD and GAD fusion proteins for yeast two-hybrid analysis, as described in [23]. The compound 3-amino-1,2,4-triazole (3-AT, Sigma-Aldrich, Inc., St. Louis, MO, USA), an inhibitor of His3, was added to CSM medium when required.

### 2.3. Serial Dilution of Cells for Growth Phenotype Analysis

Yeast strains were freshly grown on YPD or YNBcasD (for the selection of plasmids with a *URA3* selection maker) solid medium at 30 °C for 2–3 days. The cells were transferred from plates into sterile water and diluted to the same starting OD_600_ of 0.1. Five-fold serial dilutions were made and spotted on YPD plate without or with caffeine or sorbitol at concentrations as indicated in the figures. The cells were grown for 2 to 5 days at 30 °C for cell growth analysis.

### 2.4. Cellular Extract Preparation, Phosphatase Treatment, Immunoprecipitation, and Immunoblotting

The yeast strains were grown for at least six generations to reach an OD_600_ of about 0.6, and total cellular proteins were prepared as described [24]. Briefly, we mixed 1 mL cell culture with 160 μL freshly prepared solution of 1.85 N NaOH and 7.5% β-mercaptoethanol, which promoted cell lysis. Trichloroacetic acid was then added at the final concentration of 6.7% to facilitate protein precipitation. Protein pellets were recovered by centrifugation at 21,000× *g* for 5 min. Phosphatase treatment of total cellular proteins was conducted as described in [23]. For co-immunoprecipitation, cellular lysates were prepared in IP buffer (50 mM Tris-HCl, pH 7.6, 100 mM NaCl, 0.1% Triton X-100, 1 mM PMSF and 10 μM leupeptin (Sigma-Aldrich, Inc.)). Cell extracts (~2 mg proteins) were incubated at 4 °C for 1 h with 10 μg anti-c-myc antibody (9E10, Roche Diagnostics, Indianapolis, IN, USA). A volume of 30 μL washed protein G-Sepharose (Roche Diagnostics) was then added, and the samples were further incubated at 4 °C overnight with agitation. Immunoprecipitates bound to the Sepharose beads were precipitated by centrifugation at 1000× *g* for 1 min, washed three times in 1 mL TBS-T buffer each (20 mM Tris-HCl, pH 7.6, 140 mM NaCl, 0.1% Tween-20), and released by boiling in 80 μL SDS-PAGE sample buffer. Immunoblotting was conducted as described in [20]. The following antibodies were used in this study: anti-Pgk1 antibody (1:2000), rabbit polyclonal antibodies against recombinant yeast phosphoglycerate kinase generated by the lab; anti-myc antibody, clone 9E10 (1:1000, Roche Diagnostics, Indianapolis, IN, USA); anti-HA antibody, clone 3F10 (1:2000, Roche Diagnostics); HRP-conjugated goat anti-mouse antibody (1:3000, catalog # 115-035-003); HRP-conjugated goat anti-rabbit antibody (1:3000, catalog # 111-035-003); HRP-conjugated goat anti-mouse antibody, light chain specific (1:3000, catalog # 115-035-174); and HRP-conjugated goat anti-rat antibody (1:3000, catalog # 112-035-003). All four HRP-conjugated antibodies were purchased from Jackson ImmunoResearch Laboratories, West Grove, PA, USA. Chemiluminescence images of Western blots were captured and processed as described in [20].

## 3. Results

### 3.1. Hrr25 Interacts with Pin4 and Is Required for Pin4 Phosphorylation

Pin4 is a phosphoprotein that becomes hyperphosphorylated in response to DNA damage [13,25]. Two DNA damage checkpoint kinases, Mec1 and Tel1, have been reported to be required for the DNA damage-induced phosphorylation of Pin4. It is unclear what kinase is responsible for Pin4 phosphorylation under normal growth conditions. Pin4 has been reported to interact with Hrr25 in large-scale protein complex pull-down assays and yeast two-hybrid analysis. However, the possibility of Pin4 as an Hrr25 target has not been characterized systematically.

We first wanted to confirm the interaction between Pin4 and Hrr25 using a co-immunoprecipitation assay. To that end, we constructed plasmids encoding C-terminal 3xmyc-tagged Hrr25 (Hrr25-myc) and 3xHA-tagged Pin4 (Pin4-HA). Cell lysates were prepared in strains coexpressing Hrr25-myc and Pin4-HA, or expressing tagged Pin4-HA alone, and subjected to immunoprecipitation with an anti-myc antibody. Immunoprecipitates were examined for Myc- and HA-tagged proteins using Western blotting. Figure 1A shows that Pin4-HA was recovered in the immunoprecipitates from the strain expressing Hrr25-myc, but not from the strain expressing non-tagged Hrr25. Thus, we confirmed a physical interaction between Pin4 and Hrr25. Figure 1A additionally shows that the slower mobility forms of Pin4 were preferentially recovered in the Hrr25-myc immunoprecipitates, suggesting that Hrr25 has a higher affinity for hyperphosphorylated forms of Pin4 than hypophosphorylated Pin4. It has been suggested previously that Elp1 phosphorylation at serine residues 1198 and 1202 by Hrr25 stabilizes its interaction with Hrr25 [26]. The significance of a stable association of Hrr25 with its target following phosphorylation is not yet clear. It is also possible that Hrr25 may protect bound Pin4 from being dephosphorylated during immunoprecipitation.

**Figure 1 genes-16-00094-f001:**
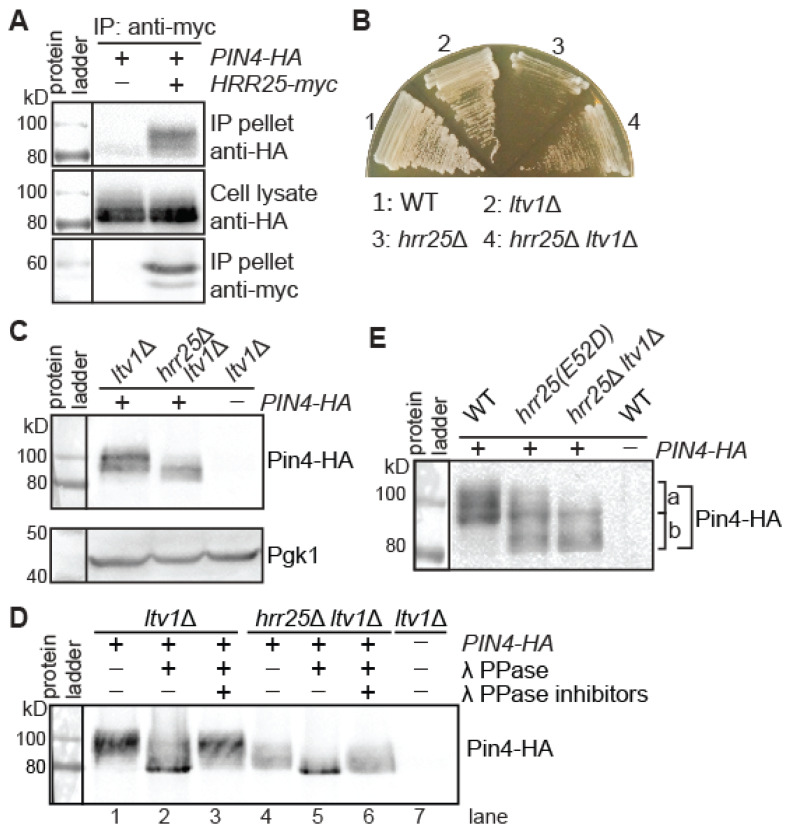
Hrr25 interacts with Pin4 and is required for Pin4 phosphorylation in vivo. (**A**) A co-immunoprecipitation assay of the interaction between Hrr25 and Pin4. Yeast cells expressing *PIN4-HA* and *HRR25-myc*, as indicated, were grown in YNBcas5D medium to the mid-log phase. Cell lysates were prepared and Hrr25-myc was immunoprecipitated as described in the Materials and Methods section. HA- and myc-tagged proteins were detected by immunoblotting. The result was representative of two independent experiments. (**B**) *ltv1*Δ suppresses the severe growth defects of *hrr25*Δ mutant cells. Wild-type (BY4741) and isogenic mutant strains (*ltv1*Δ, ZLY3475; *hrr25*Δ, ZLY4501; *hrr25*Δ *ltv1*Δ, ZLY5801) were grown on YPD medium. The picture was taken after two days. (**C**) Hrr25 is required for Pin4 phosphorylation in vivo. *ltv1*Δ and *hrr25*Δ *ltv1*Δ mutant strains expressing *PIN4-HA* were grown in YNBcasD medium. Cell lysates were prepared, and Pin4-HA was detected by immunoblotting. The result was representative of two independent experiments. Pgk1—phosphoglycerate kinase—was included as the loading control. The result was representative of two independent experiments. (**D**) Phosphatase treatment of Pin4-HA from *ltv1*Δ and *ltv1*Δ *hrr25*Δ mutant cells results in bands of the same minimal size on the immunoblot. Total cellular proteins were prepared and treated with lambda protein phosphatase (λ PPase) in the presence or absence of phosphatase inhibitors, as indicated. Pin4-HA was detected by immunoblotting. (**E**) An *hrr25(E52D)* missense mutation partially mimics *hrr25*Δ in reducing Pin4 phosphorylation. Pin4-HA was detected using immunoblotting. “a” and “b” indicate the range of phosphorylated Pin4-HA species from wild-type and *hrr25*Δ *ltv1*Δ mutant cells, respectively. The result was representative of two independent experiments. The data in panel (**C**) can also be viewed as a partial replicate of panel 1E. *hrr25(E52D)*, ZLY4467.

We next tested whether Hrr25 is required for Pin4 phosphorylation by comparing its phosphorylation state in wild-type versus *hrr25*Δ mutant cells. *hrr25*Δ leads to severe growth defects [21,27], which can be partially suppressed by an *ltv1*Δ mutation [28]. Accordingly, we generated an *hrr25*Δ *ltv1*Δ double mutant, which had better growth than an *hrr25*Δ mutant (Figure 1B), consistent with published findings. We then introduced the plasmid encoding *PIN4-HA* into this mutant and an *ltv1*Δ strain. Transformants were grown in YNBcasD medium to reach the mid-logarithmic phase. Total cellular proteins were prepared, separated by SDS-PAGE, and probed with anti-HA antibody using immunoblotting. Figure 1C shows that there was a clear mobility shift of Pin4-HA from slower to faster mobility forms in the *hrr25*Δ *ltv1*Δ double mutant compared to the *ltv1*Δ mutant, suggesting that Hrr25 is required for optimal Pin4 phosphorylation.

Another explanation for the difference in Pin4 mobility forms shown in Figure 1C is that there is proteolytic removal of ~10 kD from the N-terminal end of Pin4-HA in *hrr25*Δ *ltv1*Δ mutant cells. To differentiate these two possibilities, we treated total cellular proteins from *ltv1*Δ and *hrr25*Δ *ltv1*Δ cells expressing *PIN4-HA* with lambda protein phosphatase in the presence or absence of phosphatase inhibitors and examined Pin4 mobility by immunoblotting. When treated with lambda protein phosphatase, the broad Pin4 band seen in the immunoblot from both *ltv1*Δ and *hrr25*Δ *ltv1*Δ strains shifted downwards and a dominant, fast-migrating species appeared (Figure 1D, compare lane 2 to 1 and 5 to 4). However, in the presence of phosphatase inhibitors, the effect was largely blocked, consistent with the notion that Pin4 is a hyperphosphorylated protein. Importantly, phosphatase treatment of Pin4-HA from *ltv1*Δ and *hrr25*Δ *ltv1*Δ cells generated the fastest mobility forms of the same size, indicating that the Pin4 mobility shift caused by *hrr25*Δ in Figure 1C is due to reduced phosphorylation. The fastest mobility form of Pin4 in lanes 2 and 5 shown in Figure 1D is ~80 kD, which agrees with the predicted size of unmodified Pin4-HA, 78 kD. Together, these data indicate that Hrr25 is required for optimal Pin4 phosphorylation.

Pin4 mutations lead to synthetic growth defects with *slt2*Δ and *bck1*Δ [13]. Our findings that Hrr25 is required for Pin4 phosphorylation suggest that Hrr25 may also be involved in cell wall integrity signaling. We wanted to test whether *hrr25* mutations led to synthetic growth defects with *slt2*Δ and *bck1*Δ. Although *hrr25*Δ *ltv1*Δ mutant cells allowed us to examine Pin4 phosphorylation in the complete absence of Hrr25, the double mutant still had relatively strong growth defects (Figure 1B), which may pose problems in detecting potential synthetic growth defects in combination with *slt2*Δ and *bck1*Δ. We recently employed a missense *hrr25* mutation, *hrr25(E52D)*, to uncover two novel roles for Hrr25, namely as a negative regulator of Haa1 in the weak acid stress response pathway and as a negative regulator of Puf3 in the mitochondrial biogenesis pathway [21,23]. An *hrr25(E52D)* mutation only leads to a mild growth defect. Before we used the *hrr25(E52D)* mutation for cell growth analysis in combination with mutations in genes involved in cell wall integrity signaling, we examined its effect on Pin4 phosphorylation. Figure 1E shows that the dominant species of Pin4-HA range in ~95–110 kD in wild-type cells and ~82–97 kD in *hrr25*Δ *ltv1*Δ mutant cells. In *hrr25(E52D)* mutant cells, Pin4-HA migrates over a broader range, in sizes of ~82–108 kD. Importantly, the Pin4 migration pattern on the immunoblot from an *hrr25(E52D)* mutant is more similar to an *hrr25*Δ *ltv1*Δ mutant than to wild type, suggesting that the *hrr25(E52D)* mutation might phenocopy *hrr25*Δ in altering CWI signaling, if any. In the next three sections, we present data on cell growth phenotypes due to an *hrr25(E52D)* mutation in combination with *bck1*Δ, *slt2*Δ, and *rlm1*Δ, respectively.

### 3.2. An hrr25(E52D) Mutation and bck1*Δ* Lead to Synthetic Growth Defects

Traven et al. have described strong synthetic growth defects of *pin4*Δ with mutations in two genes encoding components of the CWI MAP kinase cascade, namely Bck1 and Slt2 [13]. Given the physical and biochemical interactions between Hrr25 and Pin4, we hypothesized that Hrr25 may regulate the activity of Pin4 in the CWI pathway. To test this possibility, we first determined whether an *hrr25(E52D)* mutation and *bck1*Δ led to synthetic growth defects. Accordingly, a haploid *bck1*Δ strain was crossed to an *hrr25*Δ strain carrying a plasmid-borne *hrr25(E52D)* mutant allele and a *pin4*Δ mutant as a control. The resulting diploid strains were induced to undergo meiosis and tetrad dissections were performed (Figure 2A,B).

**Figure 2 genes-16-00094-f002:**
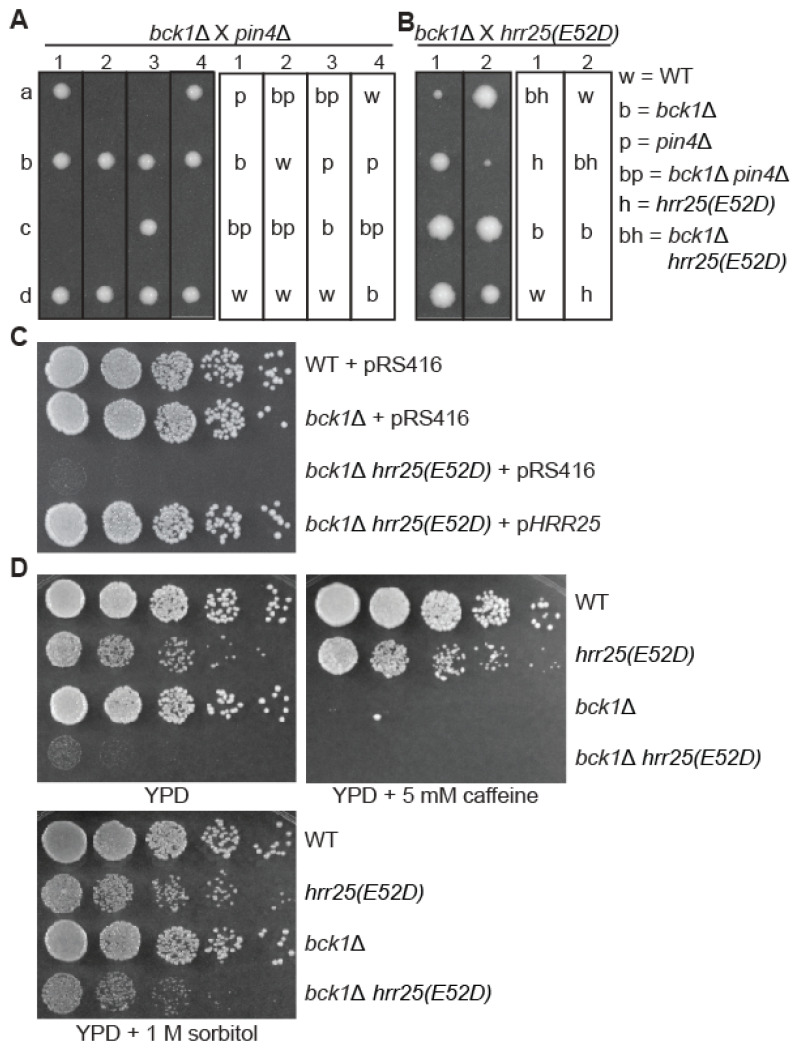
An *hrr25(E52D)* mutation leads to synthetic growth defects with *bck1*Δ. (**A**,**B**) Tetrad analysis of *bck1*Δ crossed with *pin4*Δ (**A**) or with *hrr25(E52D)* (**B**). a, b, c, and d denote the four spores from a single tetrad dissected on YPD plate and numbers indicate different tetrads. (**C**) Plasmid-borne *HRR25* rescues the severe growth defects of *bck1*Δ *hrr25(E52D)* double mutant cells. Wild type (BY4741) and isogenic mutant strains (*bck1*Δ, ZLY3512; *bck1*Δ *hrr25(E52D)*, ABY362) carrying empty vector pRS416 or a plasmid encoding *HRR25* (pMB378) were serially diluted and spotted on YPD plate. The data were representative of the growth phenotype of three independently generated *bck1*Δ *hrr25(E52D)* mutants. (**D**) 1 M Sorbitol in the growth medium suppresses the growth defects of *bck1*Δ *hrr25(E52D)* mutant cells. Wild type and mutant strains as indicated were serially diluted and spotted on YPD plates without or with the supplementation of 5 mM caffeine or 1 M sorbitol. *hrr25(E52D)*, ZLY4467. The data were representative of the growth phenotype of two independently generated *bck1*Δ *hrr25(E52D)* mutants.

In the W303 background, *bck1*Δ *pin4*Δ did not grow into visible colonies. In the BY4741 background, random spore analysis showed that *bck1*Δ *pin4*Δ mutants were able to form slow-growing colonies of variable sizes [13]. Traven et al. proposed that extragenic suppressor mutations might explain the colony size heterogeneity among the *bck1*Δ *pin4*Δ double mutant cells. During tetrad analysis, we found that nine *bck1*Δ *pin4*Δ double mutant spores failed to grow into visible colonies, which contained from ~9 to 80 cells (Figure 2A). Three *bck1*Δ *pin4*Δ double mutant progenies grew into tiny colonies. We propose that *bck1*Δ and *pin4*Δ are also synthetic lethal in the BY4741 background and that the growth defect can be suppressed by high-frequency spontaneous suppressor mutations. Unlike the *bck1*Δ *pin4*Δ double mutants, *bck1*Δ *hrr25(E52D)* double mutant cells were easily obtained but grew very slowly on dextrose medium (Figure 2B). Since *bck1*Δ and *hrr25(E52D)* single mutants resulted in no or mild growth defects, the growth phenotype of *bck1*Δ *hrr25(E52D)* double mutant cells in Figure 2B indicates that *bck1*Δ and the *hrr25(E52D)* mutation result in strong synthetic growth defects. To confirm that the growth defect is caused by the double mutation, we introduced a plasmid encoding wild-type *HRR25* into the double mutants and found that transformants grew as if they were wild type (Figure 2C). The strong growth defect caused by either *pin4*Δ or an *hrr25(E52D)* mutation in the *bck1*Δ background suggests that Hrr25 may be a positive regulator of Pin4.

Deletion mutations in *BCK1* and *PIN4* lead to compromised cell growth on medium containing cell wall stressors, such as caffeine [9,13]. Conversely, under conditions of compromised CWI caused by a *bck1*Δ or *slt2*Δ mutation, growth defects can be suppressed by an osmotic stabilizer, such as sorbitol [29]. We tested the growth of wild type, *hrr25(E52D)*, *bck1*Δ, *bck1*Δ *hrr25(E52D)* mutant cells on YPD plate without or with 5 mM caffeine or 1 M sorbitol. Figure 2D shows that 1 M sorbitol largely suppressed the growth defect of *bck1*Δ *hrr25(E52D)* mutant cells, suggesting that Hrr25 is involved in CWI signaling. Consistent with our expectations, *bck1*Δ and *bck1*Δ *hrr25(E52D)* mutants were sensitive to caffeine. *pin4*Δ results in caffeine sensitivity [13]. Since both *pin4*Δ and an *hrr25(E52D)* mutation exhibit severe synthetic growth defects in combination with *bck1*Δ, we predicted that the *hrr25(E52D)* mutation would lead to caffeine sensitivity. Paradoxically, the *hrr25(E52D)* mutation did not show increased sensitivity to caffeine (Figure 2D).

### 3.3. An hrr25(E52D) Mutation and slt2*Δ* Lead to Synthetic Growth Defects

Similarly, we examined the growth phenotypes of *slt2*Δ *pin4*Δ and *slt2*Δ *hrr25(E52D)* double mutant cells. Both *slt2*Δ *pin4*Δ and *slt2*Δ *hrr25(E52D)* double mutants generated via tetrad analysis exhibited severe growth defects on YPD medium (Figure 3A,B). Since Travern et al. have reported strong growth defects of *slt2*Δ *pin4*Δ double mutant cells [13], we focused our attention on the growth phenotype of *slt2*Δ *hrr25(E52D)* double mutant cells. To confirm that *slt2*Δ and the *hrr25(E52D)* mutation led to this synthetic growth defect, we introduced a plasmid encoding *SLT2* into *slt2*Δ *hrr25(E52D)* double mutant cells and found that the transformants had a similar growth phenotype to that of *hrr25(E52D)* mutant cells (Figure 3C). This result further strengthens the notion that Hrr25 is a novel factor in CWI signaling, possibly as a positive regulator of Pin4. Consistently, caffeine exacerbated the growth defect of *slt2*Δ *pin4*Δ and *slt2*Δ *hrr25(E52D)* double mutant strains, while sorbitol suppressed it (Figure 3D). Figure 3D also shows that *slt2*Δ led to stronger caffeine sensitivity than *pin4*Δ.

**Figure 3 genes-16-00094-f003:**
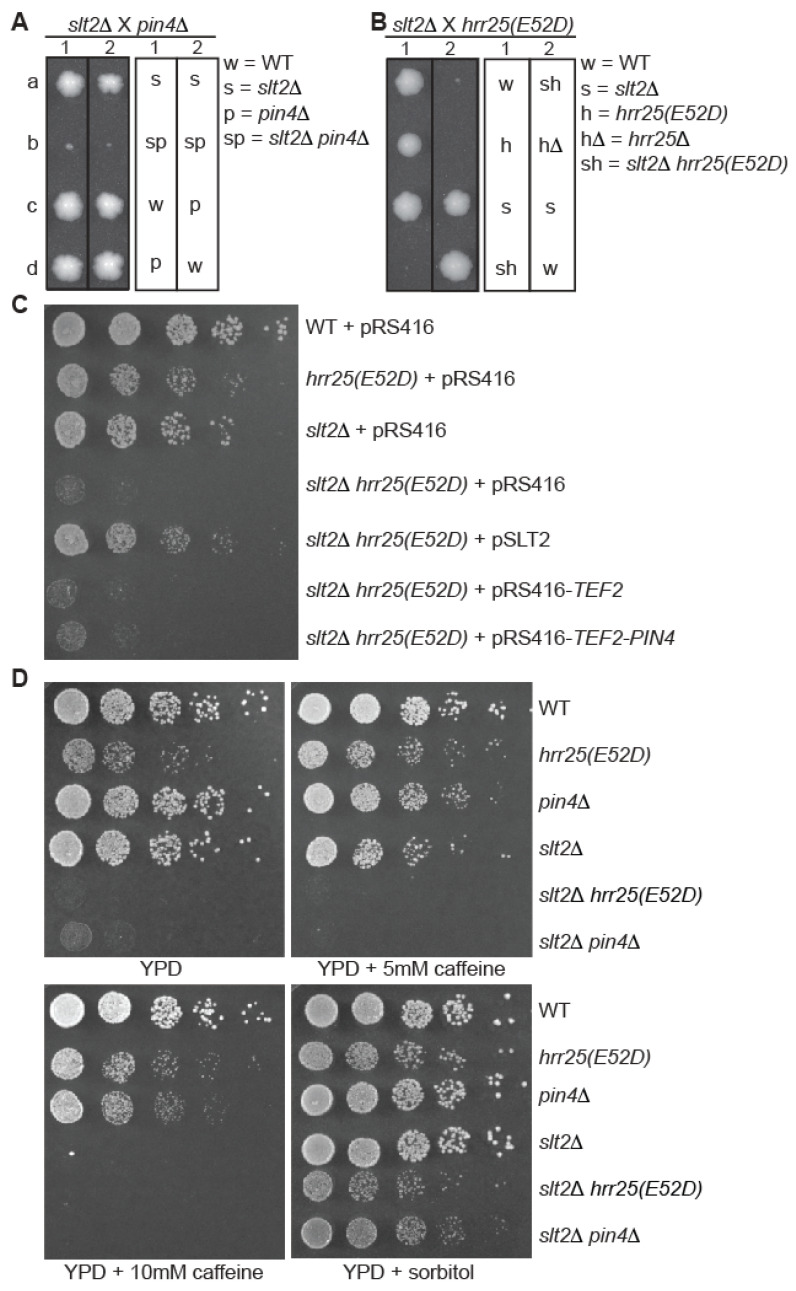
Mutations in *PIN4* and *HRR25* lead to synthetic growth defects with *slt2*Δ. (**A**,**B**) Tetrad analysis of *slt2*Δ crossed with *pin4*Δ (**A**) or with *hrr25(E52D)* (**B**). a, b, c, and d denote the four spores from a single tetrad dissected on YPD plates, and numbers indicate different tetrads. (**C**) The severe growth defects of *slt2*Δ *hrr25(E52D)* double mutant cells are suppressed by plasmid-borne *SLT2* (pAB155) but not by the expression of *PIN4* under the control of the strong *TEF2* promoter (pAB141). Wild-type (BY4741) and mutant strains (*slt2*Δ, ABY235; *slt2*Δ *hrr25(E52D)*, ABY244) carrying a plasmid, as indicated, were grown on YNBcasD plate. The data were representative of the growth phenotype of two independently generated *slt2*Δ *hrr25(E52D)* mutants. (**D**) Sorbitol suppresses the growth defects of *slt2*Δ *hrr25(E52D)* and *slt2*Δ *pin4*Δ mutant cells, while caffeine exacerbates them. Wild type and mutant strains as indicated were serially diluted and spotted on YPD plates without or with the supplementation of caffeine or 1 M sorbitol. The data were representative of the growth phenotypes of two independently generated *slt2*Δ *hrr25(E52D)* and *slt2*Δ *pin4*Δ mutants.

Together, our data on synthetic growth defects caused by mutations in *HRR25/PIN4* and *BCK1/SLT2* suggest that Hrr25 and Pin4 mediate cell wall integrity by functioning in a parallel pathway to the classic CWI signaling pathway.

### 3.4. rlm1*Δ* Leads to Increased Sensitivity to Caffeine in Combination with pin4*Δ*, but Not with an hrr25(E52D) Mutation

Rlm1 is a transcriptional activator downstream of Slt2. Mutations in *RLM1* lead to milder cell wall integrity defects compared to mutations in genes encoding the kinases of the MAP kinase cascade. This is partly because Slt2 and its pseudokinase paralog, Mlp1, regulate another transcription factor Swi4/6, which is also involved in cell wall integrity maintenance. The data presented in Figure 2 and Figure 3 suggest that Hrr25 and Pin4 may function in a parallel pathway to CWI signaling. Given that Rlm1 is positioned at a branch point of CWI signaling, it would be interesting to examine the growth phenotypes of *pin4*Δ *rlm1*Δ and *hrr25(E52D) rlm1*Δ mutant strains.

To that end, we generated *rlm1*Δ *pin4*Δ and *rlm1*Δ *hrr25(E52D)* double mutants via tetrad analysis and/or transforming yeast strains with gene knockout cassettes. Figure 4A shows that the four *pin4*Δ *rlm1*Δ mutant colonies are marginally smaller than those of wild type, *pin4*Δ single, and *rlm1*Δ single mutants, suggesting that *pin4*Δ and *rlm1*Δ lead to a very subtle synthetic growth defect. We failed to detect any synthetic growth defect between *rlm1*Δ and an *hrr25(E52D)* mutation (Figure 4B). In the presence of 10 mM caffeine, *pin4*Δ mutant cells showed clear growth inhibition when compared to wild-type cells, while *rlm1*Δ exhibited milder growth inhibition (Figure 4C). The growth of *pin4*Δ *rlm1*Δ mutant cells was almost completely inhibited by 10 mM caffeine, indicating that *rlm1*Δ and *pin4*Δ lead to severe synthetic growth defects when cell wall integrity is compromised. In contrast, *rlm1*Δ and an *hrr25*Δ mutation did not lead to a synthetic growth defect in the presence of the maximal concentration of caffeine we tested, which was 10 mM. Based on the cell growth phenotypes presented in Figure 2D, Figure 3D, and Figure 4C,D, we were able to rank the caffeine sensitivity of the single mutants as follows: *bck1*Δ > *slt2*Δ > *pin4*Δ > *rlm1*Δ. On the other hand, *hrr25(E52D)* cells did not show increased caffeine sensitivity when compared to wild type.

**Figure 4 genes-16-00094-f004:**
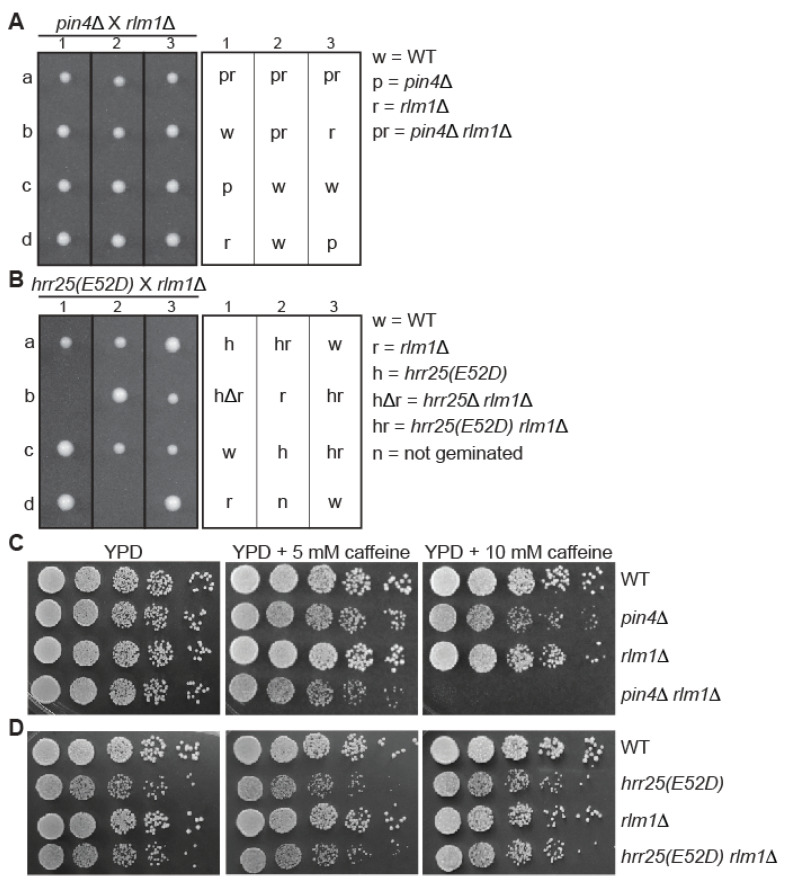
*rlm1*Δ leads to increased sensitivity to caffeine in combination with *pin4*Δ, but not with an *hrr25(E52D)* mutation. (**A**,**B**) Tetrad analysis of *rlm1*Δ crossed with *pin4*Δ (**A**) or with *hrr25(E52D)* (**B**). a, b, c, and d denote the four spores from a single tetrad dissected on YPD plates and numbers indicate different tetrads. (**C**) *pin4*Δ and *rlm1*Δ lead to synthetic growth defects on YPD medium supplemented with caffeine. Wild type (BY4741) and mutant strains as indicated (*pin4*Δ, ABY220; *rlm1*Δ, ABY350; *pin4*Δ *rlm1*Δ, ABY339) were serially diluted and spotted on YPD plates without or with 5 mM or 10 mM caffeine. The data were representative of the growth phenotype of two independently generated *pin4*Δ *rlm1*Δ mutants. (**D**) *rlm1*Δ and *hrr25(E52D)* do not lead to synthetic growth defects. Wild type (BY4741) and mutant strains as indicated (*hrr25(E52D)*, ZLY4467; *rlm1*Δ, ABY401; *hrr25(E52D*) *rlm1*Δ, ABY402) were serially diluted and spotted on YPD plates without or with caffeine. The data were representative of the growth phenotype of two independently generated *hrr25(E52D*) *rlm1*Δ mutants.

Together, our data in Figure 4 indicate that *rlm1*Δ and *pin4*Δ lead to a marginal synthetic growth defect, which is exacerbated by caffeine treatment. In contrast, *rlm1*Δ and an *hrr25(E52D)* mutation do not lead to synthetic growth defects in both the absence and presence of caffeine treatment.

### 3.5. The C-Terminal Region of Hrr25 Is Required for Optimal Pin4 Interaction and Pin4 Phosphorylation

Hrr25 protein consists of three domains: an N-terminal kinase domain (amino acid residues 1–295), a middle region (amino acid residues 296–394), and a C-terminal region (amino acid residues 395–494) (Figure 5A) [27]. To gain insights into the interaction between Hrr25 and Pin4, we wanted to identify the domain of Hrr25 that is required for its association with Pin4 using a yeast two-hybrid assay. In previous studies in our lab, we generated plasmids encoding fusion proteins of the Gal4 DNA-binding domain (GBD) and Hrr25, a kinase-dead mutant Hrr25(K38A), the N-terminal kinase domain (Hrr25(N)), a middle region truncation protein (Hrr25ΔM), or a C-terminal region truncation protein (Hrr25ΔC). We generated a plasmid encoding the Gal4 activation domain (GAD) and Pin4. Plasmids encoding GBD or GBD-Hrr25 fusion proteins were introduced into the yeast two-hybrid strain AH109, while plasmids encoding GAD or GAD-Pin4 were transformed into yeast two-hybrid strain Y187. AH109 and Y187 transformants were then mated, and diploid strains were selected for the analysis of the interaction between GBD- and GAD-fusion proteins. When there is an interaction between the bait and prey proteins, Gal4 activity is reconstituted, and the expression of *ADE2* and *HIS3* reporter genes increases. This causes cells to appear less red on histidine-replete medium and enables them to grow on histidine-free medium.

Figure 5B shows that cells coexpressing GAD-Pin4 and GBD-Hrr25, GBD-Hrr25(K38A), or GBD-Hrr25ΔM appeared less red than cells coexpressing GAD and the respective GBD fusions, indicating that Hrr25 interacts with Pin4 without the need of the middle region of Hrr25. A similar conclusion can be drawn based on the expression of the *HIS3* reporter gene assayed on histidine-free CSM medium (without or with 1 mM 3-AT to inhibit the activity of His3). In contrast, cells coexpressing GAD-Pin4 and GBD-Hrr25(N) or GBD-Hrr25ΔC appeared red on CSM + histidine medium and were unable to grow on CSM medium without histidine, indicating that C-terminal P/Q-rich region of Hrr25 is important for its interaction with Pin4. Hrr25 participates in many pathways. To our knowledge, this is the first time that the C-terminal region of Hrr25 has been ascribed a function.

A co-immunoprecipitation assay was carried out to further validate the role of the C terminal region of Hrr25 in mediating the interaction between Hrr25 and Pin4. Figure 5C shows that the deletion of the C-terminal region of Hrr25 reduces its interaction with Pin4 by 11-fold. Consistently, we observed a small reduction in the level of the most phosphorylated forms of Pin4 (Figure 5D). Together, our data indicate that the C-terminal P/Q-rich region of Hrr25 is required for both Pin4 interaction and optimal Pin4 phosphorylation.

**Figure 5 genes-16-00094-f005:**
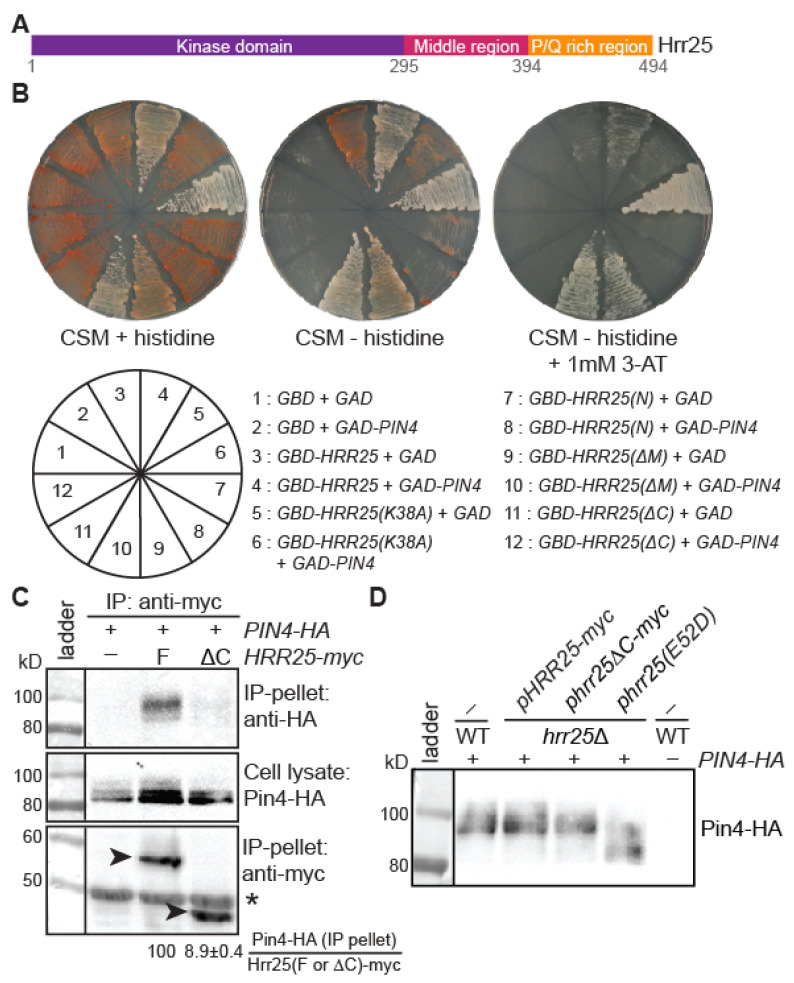
The C-terminal P/Q-rich region of Hrr25 is required for optimal Pin4 interaction and Pin4 phosphorylation. (**A**) A diagrammatic representation of the domains of Hrr25. (**B**) Yeast two-hybrid analysis of the interaction between Pin4 and Hrr25, Hrr25(K38A), or Hrr25 truncation constructs. AH109 cells carrying plasmids encoding the Gal4 DNA-binding domain (GBD), GBD-Hrr25, GBD-Hrr25(K38A), GBD-Hrr25(N), Hrr25ΔM, or Hrr25ΔC and Y187 cells carrying plasmids encoding the Gal4 transcriptional activation domain (GAD) or GAD-Pin4 were mated, and the resulting diploid cells were selected and streaked onto CSM medium with histidine, without histidine, or without histidine plus 1 mM 3-AT. (**C**) A co-immunoprecipitation assay shows that the C-terminal P/Q-rich region of Hrr25 is important for interaction with Pin4. Yeast cells expressing Pin4-HA and Hrr25-myc (F) or Hrr25ΔC-myc (ΔC) were grown in YNBcasD medium to the mid-logarithmic phase. Cell lysates were prepared and Hrr25-myc was immunoprecipitated as described in the Materials and Methods section. HA- and myc-tagged proteins were detected by immunoblotting. The arrowheads indicate Hrr25-myc or Hrr25ΔC-myc, and * denotes the heavy chain of the anti-myc antibody for immunoprecipitating myc-tagged Hrr25 proteins. (**D**) Deletion of the C-terminal region of Hrr25 reduces Pin4 phosphorylation. Wild type and *hrr25*Δ mutant cells carrying plasmids as indicated and a plasmid encoding *PIN4-HA* were grown in YNBcasD medium. Cell lysates were prepared, and Pin4-HA was detected by immunoblotting. The result was representative of two independent experiments.

### 3.6. Deletion of the C-Terminal Region of Hrr25 Leads to Increased Caffeine Sensitivity and Synthetic Growth Defects with bck1*Δ* and slt2*Δ*

We next investigated the importance of the C-terminal region of Hrr25 in CWI signaling by examining the cell growth phenotype of *hrr25*Δ*C* mutant cells on YPD medium supplemented with caffeine. Accordingly, *hrr25*Δ cells carrying plasmids encoding *HRR25* or *hrr25*Δ*C* were serially diluted and spotted onto YPD plates supplemented with various concentrations of caffeine (Figure 6A). On YPD medium without caffeine, the *hrr25*Δ*C* mutation had little effect on cell growth. In the presence of 5 mM caffeine, the *hrr25*Δ*C* mutation led to significant growth defects. In the presence of 10 mM caffeine, the growth defect due to the *hrr25*Δ*C* mutation was even greater, which was comparable to that caused by *pin4*Δ (compare Figure 6A to Figure 3D and Figure 4C). Similar to the *hrr25(E52D)* mutation, *hrr25*Δ did not lead to increased caffeine sensitivity (Figure 2D, Figure 3D, Figure 4D, and Figure 6A).

**Figure 6 genes-16-00094-f006:**
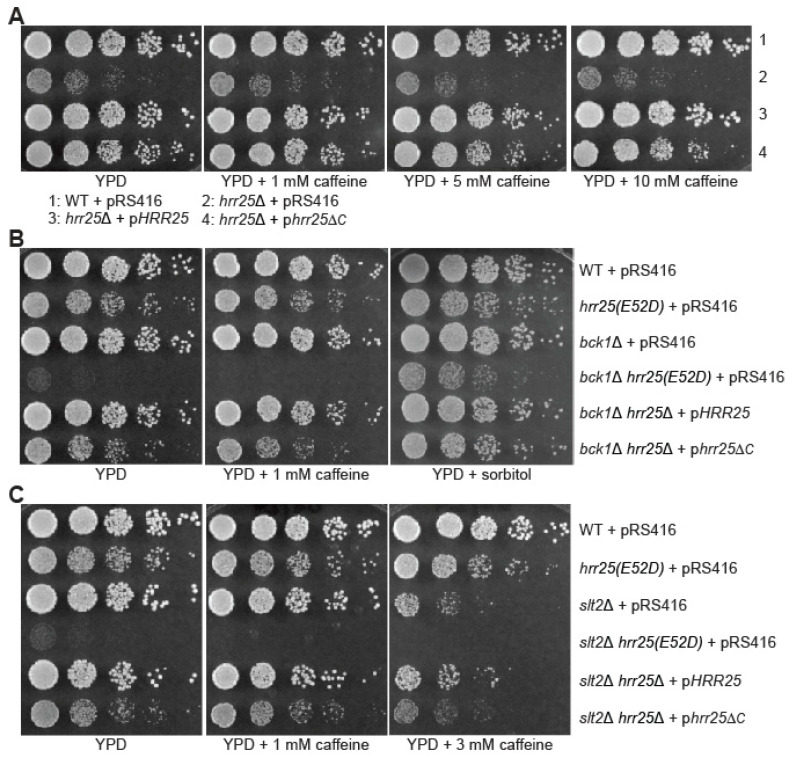
Deletion of the C-terminal region of Hrr25 leads to caffeine sensitivity (**A**) and synthetic growth defects with *bck1*Δ (**B**) or *slt2*Δ (**C**). Wild-type and mutant strains as indicated carrying pRS416, pRS416-HRR25 (pHRR25), or pRS416-hrr25ΔC (phrr25ΔC) were serially diluted and spotted on YPD plates without or with caffeine or sorbitol.

To further confirm a role for C-terminal region of Hrr25 in CWI signaling, we generated *bck1*Δ *hrr25*Δ*C* and *slt2*Δ *hrr25*Δ*C* mutant strains and examined their growth phenotype. We found that the *hrr25*Δ*C* mutation led to strong synthetic growth defects with both *bck1*Δ and *slt2*Δ (compare the last two rows of cells on YPD plates in Figure 6B/Figure 6C to Figure 6A). Significantly, 1 M sorbitol effectively suppressed the growth defect of *bck1*Δ *hrr25*Δ*C* mutant cells (compare the last two rows of cells on YPD + sorbitol in Figure 6B to YPD plate in Figure 5A). Together, our data indicate that the C-terminal region of Hrr25 is important for Hrr25’s role in CWI signaling.

## 4. Discussion

Pin4 is a phosphoprotein that has been implicated in DNA damage response and the cell wall integrity pathway. In this report, we show that Hrr25 interacts with Pin4 and is required for Pin4 phosphorylation under normal growth conditions. Based on genetic interactions presented herein and reported previously, we propose that Hrr25 is a positive regulator of Pin4 and is a novel component of cell wall integrity signaling (Figure 7). *PIN4* and *HRR25* form multiple genetic interactions with genes encoding protein kinases in the MAP kinase cascade and the transcription factor Rlm1. We found that the C-terminal P/Q-rich region of Hrr25 is important both for interaction with Pin4 and for cell wall integrity maintenance. We propose that this C-terminal region brings Hrr25 into its novel role in cell wall integrity signaling.

In this study, we used caffeine as a cell wall stressor. Traven et al. used caffeine and Calcofluor white and found that both inhibited the growth of *pin4*Δ mutant cells [13]. Our conclusion, that Hrr25 mediates cell wall integrity signaling, was based mainly on the observations that *hrr25(E52D)* and *hrr25*Δ*C* mutations resulted in strong synthetic growth defects with *bck1*Δ and *slt2*Δ in the absence of any cell wall stressor, which were suppressed by sorbitol. Despite the lack of use of another cell wall stressor, our conclusion should not be negatively affected.

This study uncovers a novel role for Hrr25, which already has many different cellular functions, including endocytosis, vesicular trafficking, ribosome biogenesis, microtubule assembly and spindle positioning, autophagy, meiosis, transcriptional regulation, mitochondrial biogenesis, and DNA damage response. How do Pin4 and Hrr25 mediate cell wall integrity signaling? We propose that they function in a parallel pathway to the cell wall integrity MAP kinase cascade (Figure 7). This is supported by the observations that *pin4 bck1*, *pin4 slt2*, *hrr25(E52D) bck1*, and *hrr25(E52D) slt2* double mutants exhibit severe growth defects, which can be suppressed by sorbitol, while single mutant strains have no growth defects (*bck1*, *slt2*, and *pin4*) or mild growth defects (*hrr25(E52D)*) when grown on YPD medium (Figure 7). Pkc1 acts upstream of the MAP kinase cascade. Unlike mutations in genes encoding Bck1, Mkk1/2, and Slt2, deletion of *PKC1* is lethal, which can be suppressed by sorbitol [31,32]. Pkc1 must regulate other proteins that are important for cell wall integrity maintenance and act in parallel to the MAP kinase cascade. Since *pkc1*, *pin4 slt2*, *hrr25(E52D) bck1*, and *hrr25 (E52D) slt2* mutants have similar growth phenotypes, it is likely that Hrr25/Pin4 may be the long-sought-after factors downstream of Pkc1 (Figure 7). Hrr25 has been reported to interact with and phosphorylate Swi6 of the SBF complex in response to DNA damage [30]. Slt2 and its paralog Mlp1 are also known to play a role in the regulation of Swi4/6 in the cell wall integrity pathway (Figure 7). Thus, another possibility is that Pin4 and Hrr25 mediate cell wall integrity maintenance via Swi4/6.

How do Pin4 and Hrr25 mediate cell wall integrity? Between Pin4 and Hrr25, Pin4 is likely to be the key regulator of cell wall integrity signaling. This notion is supported by the observation that *pin4*Δ leads to stronger growth defects than an *hrr25(E52D)* mutation when combined with mutations in *BCK1* and *RLM1*. However, this possibility is complicated by the fact that the *hrr25(E52D)* mutation only leads to a partial loss of function of Hrr25. Among the diverse functions of Hrr25, Pin4 also shares a similar role in DNA damage response [25,27,30,33]. Interestingly, a recent report by Liu et al. suggests that Pkc1 also plays a role in DNA damage response, downstream of Hrr25. Thus, there appears to be crosstalk between CWI signaling and the DNA damage response. A key question that remains to be addressed is the exact biological function of Pin4. An *hrr25(E52D)* mutation leads to defects in mitochondrial biogenesis and respiratory deficiency. However, we found that *pin4*Δ did not lead to a defect in respiratory metabolism. Thus, it is unlikely that the main biological function of Pin4 is to activate Hrr25. Both *pin4*Δ and an *hrr25(T176I)* mutation suppress the temperature-sensitive growth phenotype of a *sec12-4* mutant. However, it remains unclear how Sec12, Hrr25, and Pin4 biologically interact with each other. Whether Pin4 is involved in other Hrr25-dependent cellular processes remains to be determined.

Hrr25 is localized to different cellular sites, including P-bodies, spindle pole bodies, bud neck, endocytic sites, and the nucleus [16,34,35]. Hrr25 consists of an N-terminal protein kinase domain, a middle region, and a C-terminal P/Q-rich region (Figure 5). The middle region was reported to be required for Hrr25’s localization to endocytic sites, meiosis I centromeres, spindle pole bodies, and P-bodies. We recently showed that the kinase domain of Hrr25 alone is both necessary and sufficient for an interaction with the Puf3. The C-terminal P/Q-rich region of Hrr25 has not been ascribed a function until this study. Here we show that this domain is important for interaction with Pin4. This finding helped to solve a paradox that arose in this study: although the *hrr25(E52D)* mutation and *pin4*Δ lead to similar phenotypes, they differ in caffeine sensitivity. Based on the model presented in Figure 7, *hrr25* mutations are predicted to confer caffeine sensitivity. However, we were unable to observe such a phenotype: neither the *hrr25(E52D)* mutation nor *hrr25*Δ results in caffeine sensitivity (Figure 2D, Figure 3D, Figure 4D and Figure 6A). Amazingly, deletion of the C-terminal P/Q-rich domain leads to both caffeine sensitivity and synthetic growth defects with *bck1*Δ and *slt2*Δ (Figure 6). Hrr25 has many different cellular functions. One facile explanation is that other defects caused by the *hrr25(E52D)* mutation somehow suppress the caffeine sensitivity phenotype of *hrr25(E52D)* mutant cells. Future work is needed to determine the mechanism by which Hrr25 and Pin4 mediate cell wall integrity maintenance.

## 5. Conclusions

We can draw the following conclusions from this study: (1) Hrr25 interacts with Pin4 and is required for Pin4 phosphorylation; (2) Hrr25 is involved in cell wall integrity signaling; (3) The C-terminal proline/glutamine-rich region of Hrr25 is required for Pin4 interaction, Pin4 phosphorylation, and cell wall integrity maintenance.

## Figures and Tables

**Figure 7 genes-16-00094-f007:**
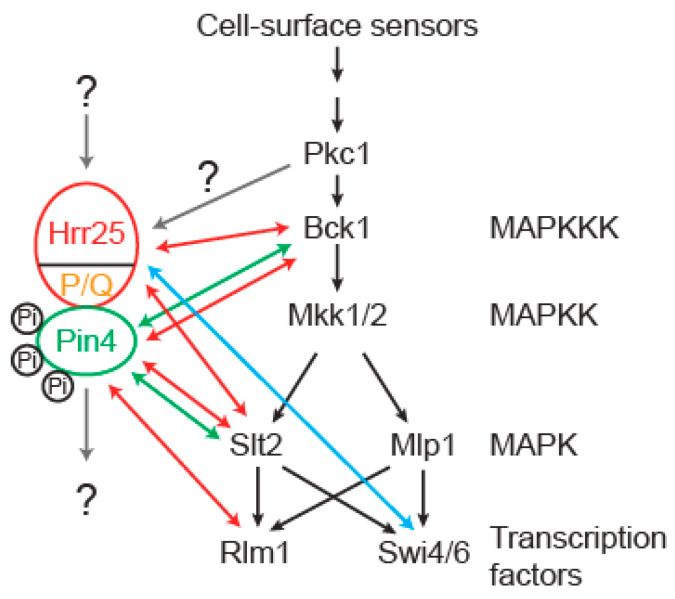
A model for the role of Hrr25 and Pin4 in cell wall integrity signaling. The classic CWI signaling pathway is adapted from the review by Levin, D. [6]. The red lines denote genetic interactions presented in this study. The green or cyan lines indicate the genetic or biochemical interactions reported previously [13,30]. P/Q, the C-terminal proline/glutamine-rich region of Hrr25. See the text for details.

**Table 1 genes-16-00094-t001:** Yeast strains used in this study.

Strain	Genotype	Source	Application (in Figure)
BY4741 (WT)	*MATa his3Δ1 leu2Δ0 met17Δ0 ura3Δ0*	Lab. stock	Figure 1A,B,E, Figure 2C,D, Figure 3C,D, Figure 4C,D, Figure 5C,D and Figure 6A–C
ZLY5753 (*HRR25-myc*)	BY4741 *hrr25Δ::kanMX4 pRS415-HRR25-myc*	This study	Figure 1A and Figure 5C,D
ZLY4501 (*hrr25*)	BY4741 *hrr25Δ::kanMX4*	[21]	Figure 1B and Figure 6A
ZLY3475 (*ltv1*)	BY4741 *ltv1Δ::kanMX4*		Figure 1B–D
ZLY5801 (*hrr25 ltv1*)	BY4741 *hrr25Δ::kanMX4 ltv1Δ::HIS3*		Figure 1B–E
ZLY4467 (*hrr25(E52D)*)	BY4741 *hrr25Δ::kanMX4 pRS415-hrr25(E52D)*	[21]	Figure 1E, Figure 2C,D, Figure 3C,D, Figure 4D, Figure 5D and Figure 6B,C
ZLY3512 (*bck1*)	BY4741 *bck1Δ::kanMX4*		Figure 2C,D and Figure 6B
ABY362 (*bck1 hrr25(E52D)*)	*MATa ura3Δ0 leu2Δ0 his3Δ1 bck1::kanMX4 hrr25Δ::kanMX4 pRS415-hrr25(E52D)*	This study	Figure 2C,D and Figure 6B
ABY235 (*slt2*)	BY4741 *slt2Δ::kanMX4*	This study	Figure 3C,D and Figure 6C
ABY244 (*slt2 hrr25(E52D)*)	BY474x *hrr25Δ::kanMX4 pRS415-hrr25(E52D) slt2Δ::kanMX4*	This study	Figure 3C,D and Figure 6C
ABY221 (*pin4*)	*MATα ura3Δ0 leu2Δ0 his3Δ1 lys2Δ0 pin4Δ::kanMX4*	This study	Figure 3D
ABY254 (*pin4 slt2*)	*MATa ura3Δ0 leu2Δ0 his3Δ1 pin4Δ::kanMX4 slt2Δ::kanMX4*	This study	Figure 3D
ABY220 (*pin4*)	BY4741 *pin4Δ::kanMX4*	This study	Figure 4C
ABY350 (*rlm1*)	*MATa his3Δ1 leu2Δ0 met17Δ0 ura3Δ0 lys2Δ0 rlm1Δ::kanMX4*	This study	Figure 4C
ABY339 (*pin4 rlm1*)	*MATa his3Δ1 leu2Δ0 ura3Δ0 pin4Δ::kanMX4 rlm1Δ::kanMX4*	This study	Figure 4C
ABY401 (*rlm1*)	BY4741 *rlm1Δ::kanMX4*	This study	Figure 4D
ABY402 (*rlm1 hrr25(E52D)*)	BY4741 *rlm1Δ::kanMX4 hrr25Δ::HIS3MX6 pRS415-hrr25(E52D)*	This study	Figure 4D
AH109	*MATa ura3-52 his3-200 trp1-901 leu2-3,112 gal4Δ* *gal80Δ* *URA3::MEL1_UAS_-MEL1_TATA_-lacZ GAL2_UAS_-GAL2_TATA_-ADE2 LYS2::GAL1_UAS_-GAL1_TATA_-HIS3*	Takara Bio USA, Inc., San Jose, CA, USA	Figure 5B
Y187	*MATα ura3-52 his3-200 ade2-101 trp1-901 leu2-3,112 gal4Δ* *gal80Δ* *met-URA3::GAL1UAS-GAL1TATA-lacZ*		Figure 5B
ZLY5767	BY4741 *hrr25Δ::kanMX4 pRS415-hrr25ΔC-myc*	This study	Figure 5C,D

**Table 2 genes-16-00094-t002:** Plasmids used in this study.

Plasmid	Description	Source	Application (in Figure)
pAB181	pRS416-PIN4-Ala10HA3, expressing *PIN4* with a C-terminal 3xHA tag.	This study	Figure 1 and Figure 5C,D
pZL3338	pRS415-HRR25-myc, expressing *HRR25* with a C-terminal 3xMyc epitope tag.	[21]	Figure 1A and Figure 5C,D
pMB378	pRS416-HRR25, expressing *HRR25* under the control of the endogenous promoter.	This study	Figure 2C and Figure 6A–C
pRS416	A centromeric plasmid with a *URA3* selection marker	ATCC	Figure 2C, Figure 3C and Figure 6A–C
pAB155	pRS416-SLT2, expressing *SLT2* under the control of its own promoter.	This study	Figure 3C
pZL632	pRS416-TEF2	[22]	Figure 3C
pAB141	pRS416-TEF2-PIN4, expressing *PIN4* under the control of *TEF2* promoter.	This study	Figure 3C
pZL3542	pRS424-ADH1-GBD, expressing the Gal4 DNA binding domain.	[21]	Figure 5B
pZL3557	pRS424-ADH1-GBD-HRR25, expressing GBD fused to Hrr25.	[21]	Figure 5B
pZL3740	pRS424-ADH1-GBD-HRR25(K38A), expressing GBD fused to a kinase-dead mutant of Hrr25, Hrr25(K38A).	[23]	Figure 5B
pZL3736	pRS424-ADH1-GBD-HRR25(N), expressing GBD fused to the N-terminal kinase domain of Hrr25, Hrr25(N).	[23]	Figure 5B
pZL3744	pRS424-ADH1-GBD-HRR25ΔC, expressing GBD fused to an Hrr25 truncation mutant without the C-terminal P/Q-rich domain.	[23]	Figure 5B
pZL3742	pRS424-ADH1-GBD-HRR25ΔM, expressing GBD fused to an Hrr25 truncation mutant without the middle region.	[23]	Figure 5B
pZL3539	pRS415-TEF2-GAD, expressing the Gal4 activation domain (GAD) under the control of the *TEF2* promoter.	[23]	Figure 5B
pAB140	pRS415-TEF2-GAD-PIN4, expressing GAD fused to Pin4.	This study	Figure 5B
pZL3701	pRS415-hrr25ΔC-myc, expressing C-terminal myc-tagged Hrr25 truncation construct without the residues 395-494.	This study	Figure 5C,D
pAB303	pRS416-hrr25ΔC, expressing nontagged Hrr25 truncation construct with the C-terminal P/Q-rich region removed.	This study	Figure 6A–C

## Data Availability

The original contributions presented in the study are included in the article, further inquiries can be directed to the corresponding author.

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
