# Peer review of "Casein Kinase I Protein Hrr25 Is Required for Pin4 Phosphorylation and Mediates Cell Wall Integrity Signaling in Saccharomyces cerevisiae"

_genes, 2025, doi:10.3390/genes16010094_

Round 1
Reviewer 1 Report
Comments and Suggestions for Authors
The manuscript entitled, Casein Kinase I Protein Hrr25 is Required for Pin phosphory- 2 lation and Mediates Cell Wall Integrity Signaling in Saccharomyces cerevisiae is a well-designed, well-organized, well-described article. The authors provided enough experimental data to prove their hypothesis. Very extraordinary work!
I would like to ask the authors to add a separate conclusion section.
iThenticate report match is 36%. Please improve that.
Reviewer 2 Report
Comments and Suggestions for Authors
The abstract provides a good overview of the research. However, consider emphasizing the novelty of the C-terminal P/Q-rich domain function more explicitly. This will help highlight the unique contribution of the study.
Here are some major comments
The description of the CWI pathway is clear but could benefit from a brief statement about its broader relevance to fungal biology or potential applications in antifungal research. Line 32.
The term “lateral inputs” is introduced but not explained fully. A brief example or clarification would help non-expert readers.
Line 49.
Consider adding details about the criteria for strain selection. This will enhance reproducibility. Line 73
The protocol for protein extraction is well-detailed but lacks a rationale for the chosen buffer composition. Briefly mentioning why these buffers were used would strengthen the section. Line 119
In line 175: The immunoprecipitation results in Figure 1A are compelling, but the potential mechanism for higher affinity toward hyperphosphorylated Pin4 could be hypothesized in more detail.
Line 245: The synthetic growth defect analysis in Figure 2 is informative but could be enhanced by a quantitative growth rate comparison.
Line 377: The yeast two-hybrid results are clear; however, a schematic representation of the interaction domains (in addition to Figure 5A) might make the data more accessible.
Line 495: The statement regarding Pin4 as the key regulator is supported but would benefit from direct experimental evidence or a plan for future studies.
Figure 7: The model could include color coding or arrows to indicate interactions newly established by this study versus previously known ones.
Reviewer 3 Report
Comments and Suggestions for Authors
The manuscript by Bhattarai et al. demonstrates that Hrr25 is involved in cell wall integrity signaling through its association with Pin4.
The manuscript is well written
I have a few comments for polishing the manuscript.
Comments
1. The authors should quantify their Western blot images.
2. In Fig. 1C and D, the authors stated that they have conducted their experiments using two biological replicates. They should conduct their experiment using at least three replicates.
3. The authors stated that Hrr25 is interacting with Pin4 protein, mediating its phosphorylation. The authors should provide insight into the specific sites undergoing phosphorylation. Did they check any additional post-translational modification?
4. Did the authors test their hypothesis using other cell wall stressors apart from caffeine?
Round 2
Reviewer 2 Report
Comments and Suggestions for Authors
The model could include color all over the contents not coding or arrows. Make figure 7 color illustrations with legends.
Author Response
Comment: The model could include color all over the contents not coding or arrows. Make figure 7 color illustrations with legends.
Response: We revised Figure 7 and its legend as suggested.
Reviewer 3 Report
Comments and Suggestions for Authors
The manuscript by Bhattarai et al. demonstrates that Hrr25 is involved in cell wall integrity signaling through its association with Pin4.
I have the following comments.
Comments
1. The authors stated that “Figure 1A additionally shows that hyperphosphorylated forms of Pin4 were preferentially recovered in the Hrr25-myc immunoprecipitates, suggesting that Hrr25 has a higher affinity for hyperphosphorylated forms of Pin4 than hypophosphorylated Pin4.” How do they prove this hypothesis? Did they use any phosphorylated antibody to check this hypothesis. They should conduct some additional experiments to validate this data.
2. The authors mentioned that “Traven et al. reported the effect of calcofluor white on pin4 mutants, which is similar to that of caffeine. However, since the phenotypes observed for hrr25 bck1, hrr25 slt2, hrr25DC bck1, and hrr25DC slt2 mutants were clear and strong even in the absence of any cell wall stressor”. They should discuss this in the discussion section.
3. The authors stated that “The data on Fig. 1C and 1E are related, with 1E repeating 1C. They can be considered additional replicates.” They should mention that in the figure legend.
